# Biomimetic Orthopedic Footwear Advanced Insole Materials to Be Used in Medical Casts for Weight-Bearing Monitoring

**DOI:** 10.3390/biomimetics8040334

**Published:** 2023-07-29

**Authors:** Sofya Rubtsova, Yaser Dahman

**Affiliations:** Department of Chemical Engineering, Toronto Metropolitan University, Toronto, ON M5B 2K3, Canada

**Keywords:** green bio-based material, polyurethane foam, poly(trimethylene ether) glycol, system castor oil, biomimetic orthopedic footwear, weight-bearing monitoring

## Abstract

Fabrication, characterization and testing of protective biomimetic orthopedic footwear advanced insole materials are introduced. The main objective of this material is to preserve and isolate a set of sensors for the Weight-Bearing Monitoring System (WBMS) device. Twenty-one samples of renewably sourced Polyurethane Foam (PUF) composed of poly(trimethylene ether) glycol (PO_3_G) and unmodified castor oil (CO) were synthesized and evaluated according to predetermined criteria. Response surface methodology of Box—Behnken design was applied to study the effect of the polyols ratio, isocyanate index (II), and blowing agent ratio on the properties (hardness, density) of PUFs. Results showed that CO/PO_3_G/Tolyene Diisocyanate (TDI) PUFs with hardness Shore A 17–22 and density of 0.19–0.25 g/cm^3^ demonstrate the required characteristics and can potentially be used as a durable and functional insole material. Phase separation studies have found the presence of well-segregated structures in PUFs having polyols ratio of CO:PO_3_G 1:3 and low II, which further explains their extraordinary elastic properties (400% elongation). Analysis of cushioning performance of PUF signified that five samples have Cushioning Energy (CE) higher than 70 N·mm and Cushioning Factor (CF) in the range of 4–8, hence are recommended for application in WBMS due to superior weight-bearing and pressure-distributing properties. Moreover, the developed formulation undergoes anaerobic soil bacterial degradation and can be categorized as a “green” bio-based material.

## 1. Introduction

Post-surgery or post-fracture recovery processes typically prescribe patients Partial Weight-Bearing (PWB) to avoid overloading. However, this is a challenging task as 10% of patients who exceed the allowable pressure limit end up with prolonged recovery times and repeated surgery procedures result in an annual indirect cost of USD 95 billion, mostly due to lost wages [1]. Therefore, clinicians and researchers will greatly benefit from a load monitoring device that can continually track the weight-bearing behavior of a patient between follow-up visits.

Today a plethora of pressure monitoring devices are available on the market (F-Scan^®^, Pedar^®^, Vicon^®^, Gait Aid^®^, etc.) However, no commercially available products are capable of recording the load placed on an injured limb for more than 1 h and are not addressed specifically to the problem of weight-bearing monitoring after surgery recovery [1]. Furthermore, they are expensive, ranging from $6000 to $16,000, making it impractical to apply these systems to large-scale clinical trials and general patient monitoring.

The WBMS consists of Force Sensing Resistors (FSRs) located on the first and fifth metatarsal heads and in the heel area. Sensors and connecting cables are sandwiched between protective biomimetic orthopedic cushioning foot insoles. Both sides of the insoles have an adhesive backing and can be firmly attached to the cotton stockinet; a cast is then applied as a normal procedure. The signal from FSRs transmits to a control pod attached to the outside of the cast and then to the patient’s smartphone. The WBM app enables the acquisition of real-time alerts (sound, vibration) when a weight threshold (20 lb) is exceeded and saves the history for doctors and long-term caregivers.

Although FSRs are often first-choice sensors due to their low cost, availability of options to create arrays of sensors and their slim shape factor, they have a low accuracy of measurement (±10%) [2] and temperature sensitivity. Besides temperature increases due to heat generation within the electric circuit, the study by Shariatmatardi et al. (2012) found that temperature inside the footwear increases by 8–13 °C after 15–20 min of activity. The economic benefits of FRSs outweigh their weaknesses; however, the design of protective pressure-distributing insoles with low thermal conductivity is necessary to overcome the FRSs’ limitations [3].

Today, a wide range of new materials have been introduced into the market, including foam rubbers, plastics, warp-knitted fabrics and cellular polymers that possess suitable properties and characteristics for use in off-loading insoles. A beneficial strategy would be to sort out the best candidates for cushioning purposes and then analyze, try to mimic and further enhance their mechanical and physical properties. However, from analysis of research studies, it is evident that most suitable materials proposed by researchers are completely dependent on the methodology (the type of tests performed, for example, compression, compression set, shear force, etc.) and the material formulation (the same material but of different density/thickness/hardness). Neither one of these parameters nor all together give a conclusive answer about which mechanical properties are responsible for improved cushioning and durability of the materials.

Although the research is not conclusive, the following trends were observed:Moderately deformable materials have the best durability and shock attenuation properties [4,5,6]. Additionally, Poron^®^ and Plastazote^®^ (both moderately deformable materials according to Campbell’s methodology) were found to significantly reduce peak pressure in biomechanical studies by Tong et al. (2010) [7] and Birke et al. (1999) [8].In the majority of research studies [3,9,10,11,12,13], PU open cell foams and PU elastomers are claimed as superior in pressure attenuation in comparison to closed cell foams, latex, felt and leather. Slobodinyuk et al. (2023) developed an efficient method for synthesizing amino-terminated oligotetramethylene oxides. The resulting products can be used as hardeners for oligomers with terminal epoxy groups synthesized from cyclic and cycloaliphatic diisocyanates [14]. These elastomers showed excellent shape memory, demonstrated by significantly high shape fixity and shape recovery ratios. Furthermore, it has a self-healing ability, which was demonstrated using the coating based on the developed polymers [14].Low-density materials tend to be less durable and have worse weight-bearing properties. Rome et al. (1991) determined that the mean density, 0.333 g/cm^3^ and hardness Shore A 32–35 is sufficient to prevent the material from quickly deforming, while on the other hand, not being too stiff [9]. Birke et al. (1999) suggested Poron^®^ materials of hardness Shore A 15–25 for the fabrication of orthotics for patients with diabetes [8]. Healy et al. (2012) reported PU foam Shore A 35 and 55, with the latter being best at significantly decreasing plantar pressure [11]. However, the review did not analyze factors that contribute to shock-absorbing performance. Certainly, it is challenging to find the link between mechanical characteristics and in-shoe performance because some parameters are impossible to simulate. However, specific standards currently exist in the footwear industry, for instance, Shoe and Allied Trades Research Association (SATRA), UK, has established a list of value ranges that correspond to a quality shock absorber. As it follows, parameter values for Poron^®^ materials are close to the optimal values developed by SATRA Technology Center [15]. Therefore, targeted characteristics that should be tailored for the quality performance of insole devices resemble properties of commercially available Poron^®^.

PWB regime is part of a recreation process for patients with lower limb fractures/strains/sprains. To avoid frequent foot overloading and achieve better patient compliance with PWB requirements, the application of the Weight-Bearing Monitoring System (WBMS) is highly attractive. However, it is challenging to find an affordable material for a protective biomimetic insole cover that is also a good shock absorber. While it is important to develop a functional and inexpensive material, it is just as important to incorporate a high percentage of renewably sourced content. PUF, composed of PO_3_G and CO, possesses the necessary hydrophobicity and mechanical properties suitable for application as an insole material [16]. CO is a plant oil that has several advantages, including easy availability, sustainability and relatively low and stable cost and is extensively used for making eco-friendly polyols in PU foam [17,18]. Among the nature-derived linear polyols, the most attractive are PO_3_G—products of corn fermentation. Moreover, polyurethane foams of vegetable origin (castor oil, palm oil, rapeseed oil) undergo biodegradation in the presence of microorganisms [19]. Based on calculations by Ogunfeyitimi et al. (2012), the price of CO-based foam is 25% lower in comparison to 100% petroleum-based counterparts with similar characteristics [20]. Taking the above-stated considerations into account, a renewably sourced PUF material was selected for synthesis and its potential for shock-absorption insole fabrication investigation.

## 2. Materials and Methods

### 2.1. Materials

All reagents used in the present research were used as received. Castor oil (chemically pure) with hydroxyl and acid values of 162 and 2 mg KOH/g, respectively, was purchased from Sigma Aldrich (St. Louis, MO, USA). Poly(trimethylene ether) glycol (PO_3_G, Velvetol 1000, Mn = 1046 g/mol) was kindly supplied by Allessa GmbH (Frankfurt, Germany). Gelling catalysts (DBTDL) and blowing catalysts (DABCO 33LV) were purchased from Sigma Aldrich, USA. TDI was used as HRs in PU synthesis and was also supplied by Sigma Aldrich. Niax L-580 silicon oil surfactant was kindly provided by Momentive Performance Materials (Waterford, NY, USA). Distilled water was used as an environmentally friendly blowing agent. Details of the raw materials used in the study are summarized in Table 1.

### 2.2. Preparation of PUFs

Preparation of polyurethane foam is a complicated chemical process that involves two types of chemical reactions: gelling (formation of a polymer) and blowing (formation of voids).

#### 2.2.1. Synthesis of Pre-Polymer

The pre-polymer method was used to prepare polyurethane [21]. The synthesis was carried out in a 250 mL glass reactor at normal pressure and agitation rate at 500 rpm (by standard tri-blade impeller) to ensure homogeneous mixing. Predetermined amounts of PO_3_G, CO and surfactant were placed in the reactor and heated to 70 °C on the oil bath under continuous agitation. Once the required temperature was achieved, a predetermined amount of TDI was added to the reaction mixture under continuous agitation. Depending on the foam’s composition, pre-polymer formation was finished after 15 ± 5 min and the reactor was removed from the oil bath. Mixing was terminated for 5 min to let excessive air escape the reaction.

#### 2.2.2. Synthesis of PUF

After mixing was terminated and the mixture was left to rest for 5 min, small amounts of DABCO, DBTDL and water were mixed separately in a bicker and injected into the reactor using a syringe. The mixture was stirred for 10 s at 500 rpm and then poured into a pre-heated Teflon-coated mold. All foams were cured in the laboratory furnace at 60 ± 1 °C for 15 ± 5 min. Foam sheets had Length/Width/Height dimensions of 22.5 cm/16.5 cm/0.6 ± 0.1 cm, respectively. The samples were stored in a closed container at 25 ± 2 °C and 65% relative humidity (rh) without exposure to light. Characterization of the samples was performed at least 48 h after synthesis.

### 2.3. Design of Experiments

BBRSM was chosen in the present study to analyze the relationship between the foam’s composition and its physical properties. The main objective of the experimental design was to develop a model to link hardness and density with the composition of the foam [22]. This study will address the effect of isocyanate levels, crosslinking and blowing agent ratio on foam density and hardness.

Factor P represents the ratio of PO_3_G to castor oil taken by equivalents. It is well known in the literature that the polyurethane foam that utilizes CO as the only OH source is very rigid due to high levels of cross-linking. On the other hand, high molecular weight linear polyols result in very soft and flexible foams. A mixture of both polyols (PO_3_G and CO) was utilized in the foam to obtain an optimal combination of these components. PO_3_G:CO ratios of 3:1; 1:1 and 1:3 were analyzed in the current study.

Factor I is the isocyanate index. This parameter was chosen for investigation because –NCO moieties take part in the reaction with water and produces CO_2_, which expands polyurethane polymers, hence creating voids. The isocyanate index was studied at three levels: 80, 100, and 120, corresponding to scarce, equilibrium and excess amounts of isocyanate, respectively. The hardness of the foam is also linked to isocyanate levels, i.e., the higher the isocyanate index, the harder the segments (HS) formed in the polymer, and hence the harder the foam.

Factor W is the blowing agent (water) ratio. Water is another component in the reaction mixture and is responsible for the formation of carbon dioxide gas. Since the density of the foam is inversely proportional to the amount of CO_2_ emitted, the ratio of the blowing agent is also expected to be inversely related to foam density. The ratios of the blowing agent were set at 0.5, 1.0 and 1.5 parts per hundred parts of polyol (pphp). Urea linkages produced as a result of the reaction between isocyanate groups and the blowing agent also affect the hardness of the polymer on a micro-level due to the fact that urea linkage is less flexible than urethane linkage and makes soft segment (SS) chain movements more restricted.

The levels of the factors were coded as −1 (low), 0 (middle) and +1 (high) and organized in a design matrix (see Table 2). Response variables were density of the foam [g/cm^3^] and hardness [Shore A units]. The Box—Behnken design included a single block of experiments where replicates were generated for the middle point (000); the rest of the design was not replicated. A total of 15 experiments were required for this design. ANOVA analysis and plotting were performed using the Minitab software. A constitutive equation was considered to forecast the area of optimal parameter values and used for planning further experiments.

### 2.4. Characterization

#### 2.4.1. Fourier-Transform Infrared Spectroscopy

A Fourier-Transform Infrared Spectroscopy (FTIR) (Cary 630 FTIR, Agilent Technologies, Waltham, MA, USA) was used to collect spectra on foam surfaces. Samples were cut into 2 cm × 2 cm × 1 cm cubes from the center of the foam buns; three samples were tested using each formulation. The foam was pressed against the ATR crystal to ensure complete contact. All data were recorded at 25 °C in the 4000–600 cm^−1^ range, at a resolution of 4 cm^−1^. The collected spectra were normalized with respect to the absorbance of the aromatic C=C stretching in TDI at 1600 cm^−1^.

#### 2.4.2. Mechanical Testing

##### Hardness

The hardness of the sheets was tested using a Shore A Digital hardness tester (HTTK-37, TekcoPlus, Hong Kong, China) based on the SATRA TM 205 standard. The test sample was prepared such that it was flat and both surfaces were smooth. Sheets were piled up so that the thickness was 11 ± 1 mm. A pre-set load was then applied and the amount by which the indenter penetrated the material was indicated on a digital gauge calibrated in Shore A hardness units.

##### Density of Insoles

The density of the insole materials was measured using the SATRA TM 12 test method. Polyurethane foam was cut into cylindrical-shaped samples with height in the direction of foam rise. The thickness of the samples was 0.55 ± 1 mm and the diameter of all samples was 20 ± 0.1 mm. Three specimens were prepared for each material type. Samples were conditioned in the vacuum furnace at 60 °C for 24 h until a constant mass was reached. The mass of the test specimen was then measured using analytical balances and recorded as [M] to the nearest 10 mg. The density of each test specimen was calculated in g/cm^3^ using Equation (1) and the density was averaged.
(1)$$Density\left(g/cm^{3}\right)=\frac{\left[M\right]\times 1000}{\left[V\right]}$$

##### Compression Set

The compression Set (CS) of the polyurethane foam was determined using the constant stress method, SATRA TM 64, at the room (23 °C) and elevated (40 °C) temperatures. For each material type, three polyurethane foam specimens (sheets 60 mm in diameter and 6 ± 1 mm in height) were cut out. These sheets were placed on the lower platen of the press so that the centers of the test specimen formed an equilateral triangle with the edges of adjacent test specimen approximately 5 mm apart and the center of the triangle aligned with the center of the platen. The initial height of the specimens was recorded at T_0_.

The spacer plate was placed on top of the test specimens. A compressive force of 453 kg-force, equivalent to a pressure of 7 kg-force/cm^2^, was applied to the test specimens using a laboratory press. The apparatus was left for 24 h without any further adjustments. The specimens were then released from the equipment and left for 1 h at room temperature. The thickness of the specimens was measured again and recorded at T_1_. CS at room temperature (denoted as CS_23_) was calculated according to Equation (2):(2)$$CS\left(\%\right)=\frac{T_{0}-H_{1}}{H_{0}}\times 100\%$$

Analogous measurements were performed at the elevated temperature. The specimens were prepared and placed on the platen as described above; the laboratory press was placed in a furnace at 40 °C for 24 h. CS at the elevated temperature (denoted as CS40) was also measured after all specimens had recovered for 1 h.

##### Water Absorption and Desorption Properties of Insole Materials

Water absorption and desorption properties of insole materials were studied according to SATRA TM 6. Three cylindrical-shaped specimens (diameter 20 mm and height 6 ± 1 mm) from each sample were cut in the direction perpendicular to foam rise; top and bottom skins were preserved. All specimens were stored at standard atmosphere (55% rh and 22 ± 2 °C) and mass [M_0_] was measured using analytical balances. Three specimens were placed in a beaker and a sufficient amount of water (T = 25 ± 2 °C) was poured into the vessel to cover the test specimens to a depth of approximately 20 mm.

All samples were removed from the water after 8.0 ± 0.1 h and the excess water was mopped off from the surface of the test specimens using filter paper. The mass of the test specimen was measured in mg using analytical balances and recorded as [M_w_] to the nearest 10 mg. To measure water desorption, wet test specimens were left to dry in air at standard atmosphere for 16.0 ± 0.2 h, after which the final mass of the specimens, [M_f_], was determined. Water absorption (WA%) and water desorption (WD%) were calculated as a percentage of the original mass of the test specimens using Equations (3) and (4):(3)$$WA\left(\%\right)=\frac{M_{w}-M_{0}}{M_{0}}\times 100\%$$
(4)$$WD\left(\%\right)=\frac{(M_{w}-M_{f})-M_{0}}{\left(M_{w}-M_{f}\right)}\times 100\%$$

##### Tensile Properties of Insole Materials

Tensile properties of insole materials were studied in compliance with the SATRA TM 2 test method (Tensile properties of insole materials). A rectangular test specimen was gradually stretched using a tensile testing machine (ELW(EX) Auto-tensile tester, Labthink Instruments Co., Jinan, China). Three test specimens with the dimensions shown in Figure 1 were cut from the foam sheet with a thickness of 6 ± 1 mm in the direction perpendicular to foam rise. The tensile testing machine was adjusted so that the jaws were 100 ± 1 mm apart. Both ends of the test specimen were inserted in the corresponding jaws of the tensile testing machine and clamped. The tensile testing machine was operated at an elongation rate of 100 ± 10 mm/min. All test specimens were tested until failure. The following test parameters were recorded for each test specimen: extension at maximum load (%), tensile strength at maximum load (MPa) and Modulus of elasticity (MPa).

##### Cushioning Properties of Insole Materials

Cushioning properties of insole materials were studied using the SATRA 159 test method. The method assesses two different cushioning properties: Cushioning Energy (CE) and Cushion Factor (CF). CE is the energy required to gradually compress a specimen of a given thickness up to standard pressure. CE is sub-divided into CE during walking (CEw) and CE during running (CEr) and is calculated as the area under the compression curve restricted by the *x*-axis and the vertical lines that correspond to the values of loading 113 N and 216 N for walking and running, respectively.

To calculate the CEw and CEr of PUF, circular specimens with diameters of 2 cm were cut out of sheets of material 6 ± 1 mm thick in the direction perpendicular to foam rise. Bottom and top skin layers were retained. The sample’s final thickness was recorded as [t]. Testing was performed using XLW(EX) Auto-tensile tester (Labthink Instruments Co., China) operated at a compression rate of 20 mm/min. The test specimen was placed between two horizontal compression surfaces so that the line of action of the force during the test passed through the center of the test specimen. The distance between the horizontal compression surfaces was adjusted until there was a gap of approximately 1 mm between the top horizontal compression surface and the upper surface of the test specimen. The tensile testing machine was operated such that the horizontal compression surfaces moved together until a force of 245 N was recorded on the test specimen. The direction of travel of the horizontal compression surfaces was reversed so that they moved apart until the force on the test specimen was reduced to 0 N. The procedure was repeated seven times and the force versus compression trace produced by the tensile testing machine was taken for the eighth time. The area under the compression curve was calculated using Origin Pro software.

CF was assessed using a test specimen with a thickness greater than 16 mm. The volume of the test specimen under no load was multiplied by the pressure on the surface of the test specimen at a predefined loading. The obtained quantity was then divided by the CE of the specimen at the predefined load; the result is CF. To determine CF, circle-shaped test specimens with a diameter of 2.85 cm were cut using a knife cutter. The thickness of foam specimens was recorded as [T] and was 6 ± 1.5 mm. The cut test specimens were stored in a standard controlled environment of 25 ± 2 °C and 65 ± 2% rh for at least 48 h prior to measurements. The procedure analogous to the determination of CE was performed on the individual specimens. CFs of the materials were calculated using Equations (5) and (6):(5)$$CFw=\frac{\left[T\right]\times 113}{\left[\mathrm{CEw}\right]}$$
(6)$$CFr=\frac{\left[T\right]\times 216}{\left[\mathrm{CEr}\right]}$$

#### 2.4.3. Differential Scanning Calorimetry

Differential scanning calorimetry (DSC) (Q1000, TA Instruments, New Castle, DE, USA) was used to determine thermal transitions within the materials. Approximately 2–3 mg of foam was loaded into an aluminium pan and sealed hermetically. The sample was first heated at 10 °C/min to 110 °C and equilibrated for 2 min before being cooled down to −80 °C. The second temperature ramp heated the samples to 250 °C at 10 °C/min and was used to determine the characteristic transitions of SSs.

The T_g_ of polyol (PO_3_G) was measured in the same instrument by loading approximately 5 mg of the polyol into an aluminium pan and sealing it hermetically. The sample was first cooled to −100 °C and equilibrated for 5 min and then heated to 240 °C at 10 °C/min. T_g_, T_c_ and T_m_ were determined using TA Analysis software.

#### 2.4.4. Biodegradation Test using Soil Burial Method

The degree and rate of aerobic biodegradation of the castor oil-based foams were determined by burying them in soil under laboratory conditions. The foams were cut into dumbbell-shaped specimens of 5 ± 0.5 mm thickness and conditioned at standard environments for 24 h; the masses of the test specimens were then recorded. Soil composition and compost were prepared in compliance with ASTM D 5988. Four soil beds were developed and four sets of three samples were buried. PUF samples were buried horizontally in wooden boxes containing 18 cm deep soils to ensure anaerobic biodegradation. Water was added daily to maintain moisture inside the chambers. Four sets of three samples were set aside as negative controls of biodegradation. The samples were stored in a dark box in a standard environment to avoid exposure to light. The samples were taken out of the container after 2 months, 3 months, and 4 months, washed thoroughly and dried at 40 °C in a furnace until a constant mass was obtained. Mass loss, spectral changes, and tensile properties of specimens were evaluated.

#### 2.4.5. Optical Microscopy Characterization

Optical microscopy images were recorded using an optical binocular microscope with an attached digital camera (Motic X/X2, Moticam, Schertz, TX, USA). For this purpose, samples were cut with a razor knife into a sheet 1 mm thick and placed on a glass slide with no further preparation. Calibration of the Motic camera was performed using calibration slides. Cell diameters of PUFs were determined using Image Plus software. The Image Plus software calculated the areas of the cell, approximating the cells to a circular shape, and automatically provided statistical distributions of cell diameters in selected regions of the foam. The diameter of the cell was then calculated from the area of the cell (assuming a circular cell shape). The experiment was repeated three times for three randomly chosen cross-sections of a PUF sample sheet.

#### 2.4.6. Scanning Electron Microscopy Characterization

A morphological study of the final polyurethane-based scaffold was conducted using a scanning electron microscope (SEM). The JEOL/OE equipment model JSM-6380 LV (Oxford Instrument, U.K.—software version SEI England) with a monochromator (Al Xray source) was operated between 5–20 kV, generating high-resolution images. Samples with dimensions of 1 cm × 1 cm × 1 cm were cut from the center of tested foams perpendicular to the growth direction and fixed in the sample holder of the microscope with no further preparation. Individual cell size was calculated using the scale bar given by the SEM image.

## 3. Results and Discussion

### 3.1. Polymerization Reaction Conditions

In this study, the FTIR-ATR was used to determine isocyanate conversion during polyurethane formation. The isocyanate absorption band is assigned at approximately 2270 cm^−1^ and the decay in intensity of this absorbance was used to monitor the isocyanate group conversion during the polymerization and is presented in Figure 2.

The kinetics of the CO/PO3G/TDI synthesis reaction is presented in Figure 3a–d. Panels a–d correspond to the different combinations of catalysts and heating regimes applied. The dotted curve in each panel demonstrates the relative change in –NCO band transmittance with respect to the methylene group signal and the solid curve indicates isocyanate conversion, *p*. The dotted curve/part of the curve was constructed based on an average decrease in the T_NCO_/T_CH2_ transmittance ratio from three replicates of the experiment. The conversion, *p*, was plotted according to Equation (7):(7)$$p=1-\frac{T_{NCO}/T_{CH2}}{{\left(T_{NCO}/T_{CH2}\right)}_{0}}$$

As shown in Figure 3, four different regimes are described in Panels a–d. Heating regiment and combination of catalysts have a significant influence on the progress of the reaction. As shown in Panel d, the best result in terms of conversion rate was achieved by applying a combination of both catalysts and post–curing of the samples.

It can be seen from Table 3 that production time for regime D plummeted in comparison to regime A. In regime D, the sample could be demolded in as little as 18 ± 5.2 min. This parameter approaches the typical time (5–20 min) required for production of custom-shaped polyurethane foam using the RIM technique [23].

Silicon oil surfactant was used in this study to increase miscibility of the components. Although the typical amount of surfactant for renewable PUF sources ranged from 1.1 pphp [16] to 18 pphp [24], the formulations in this study were tested using 1–4 pphp of silicon surfactant. The foam collapsed at 4 pphp of surfactant because the presence of surfactant interferes with the polymerization reaction. The foam with 2 pphp of surfactant was more irregular and had a combination of larger cell sizes of 400 and 500 μm. The composition containing 3 pphp of the surfactant had bell-shaped cell-diameter distributions, hence the foam was more homogeneous and primarily contained cells 200 μm in diameter. Therefore, 3 pphp of silicon surfactant was chosen as the optimal concentration and used further on.

### 3.2. Design of Experiments

The Box—Behnken experimental design for Response Surface Methodology (BB RSM) was applied for the development of the empirical model, analysis of the main effects and further prediction of PUF properties.

Response surface for the hardness of PUF as a response variable demonstrated curvature and it is for this reason that Box—Cox transformation was employed to increase the sensitivity of the model. The Pareto chart (see Figure 4) and Table 4 demonstrate significant and non-significant terms in the proposed model. After statistical analysis, the regression equation in uncoded units (only statistically significant terms included), which demonstrates the contribution of factors to the hardness property of the material, is presented below:(8)$$Hardness={\left(0.6423-0.1280\cdot P-0.1583\cdot I\right)}^{-2}$$

The proposed model has a coefficient of determination R^2^ = 92.94%, which is reasonably high and confirms that the model adequately responds to changes in the studied factors. Factor I had the highest standardized effect and hence the highest impact on the hardness of the material. The positive value of the coefficient in Factor I from the regression equation as well as the steepest slope on the plot of main effects (Figure 4) proves that isocyanate level positively contributes to the hardness of PUF. Another significant effect is *p*. The main effects plot demonstrates that the mean hardness of the material increases when the ratio of CO:PO_3_G approaches 1:3. Additionally, the *p* term of a constitutive equation had a positive value, which proves that a decrease in the CO:PO_3_G ratio has the opposite effect on hardness. Factor W (i.e., blowing agent level) had minimal effect on the hardness of PUF. This can be concluded from the Pareto chart (Figure 4) and the value of the coefficient in the constitutive equation. The two-way interactions plot is presented in Figure 4. The contribution of the interaction of all factors was insignificant since the slopes of the interaction curves were relatively small. To visualize the model, contour plots of the hardness of PUF versus a combination of factors were constructed (Figure 4). The contour plot indicated that the optimal region of the parameters was located at high levels of Factors P and I and low level of Factor W.

BBR RSM was also applied to analyze the effect of polyols ratio, blowing agent and isocyanate level on the density of PUF. Analogous to the hardness of the PUF, the response surface for the density of the PUF was highly uneven. For this reason, Box—Cox transformation based on an optimal lambda value was employed to achieve a non-significant lack-of-fit for the model. After statistical analysis, the regression equation in uncoded units (only statistically significant terms included), which demonstrates the contribution of factors to the hardness property of the material, is presented below:(9)$$Density={\left(4.554+0.783\cdot I+1.030\cdot W+0.522\cdot I\cdot I-1.306\cdot P\cdot W\right)}^{-1}$$

Initial evaluation of significant and non-significant terms was performed by analyzing the Pareto chart presented in Figure 5 and Table 5. As can be seen, Factor W had the highest value of standardized effect and the steepest slope on the main effect plot (Figure 5), hence the largest impact on the density of the PUF. Thus, the higher the level of blowing agent and II in the material, the lower the density of PUF.

Interestingly, the two-way interaction between factors P and W is another significant factor. The interaction plot of the P × W (Figure 5) effect demonstrates that the concentration of the polyols affects density to a high extent at the lowest blowing agent. In addition, the coefficient of the P × W term in the regression equation has a negative value, which means that an increase in the PO_3_G content creates lower-density foam. Lastly, the interaction between I × W has an impact only at the lowest level of the blowing agent. According to the plot (Figure 5), at a low level of blowing agent, an increase in II leads to an extremely high density of the material. The area of optimal parameter levels can be predicted by examining contour plots. Considering the analysis conducted for hardness as a response variable, the levels of either or both Factors P and I should be preferably kept at a high level.

Through analysis of residual plots, the requirement of normal error distribution is satisfied for both hardness and density models. Moreover, a random pattern created on the plot of the model’s residuals versus fitted values supported the fact that the model captures all explanatory information and responds well to changes in factors.

### 3.3. Mechanical and Physical Testing

A total of 21 PUF samples were synthesized and further tested to determine hardness, density, tensile, compression and cushioning properties. The PUF results obtained were compared with the characteristics of Poron Blue (PB) material, which is a leader among commercial shock-absorbing materials. All samples were coded with 3-digit titles, with the first digit as the level of PO_3_G:CO (0-ratio 25/75; 1-ratio 50/50; 2-ratio 75/25), the second digit as the II (0-II = 80; 1-II = 100; 2-II = 120; 3-II = 140), and the third digit as the level of blowing agent (0–0.5 pphp; 1–1 pphp; 2–1.5 pphp).

The evaluation of the hardness and density of the PUFs was performed at the first stage of testing. Hardness and density diagrams of predicted and measured values are presented in Figure 6a and Figure 6b, respectively. The area of optimal characteristics is shaded in the diagrams.

Some studies [25,26] claim that CO creates crosslinking that results in the formation of rigid foam; however, the opposite was observed in the present study. The reason for this observation could be the presence of minor traces of other fatty acids in the CO. At minor CO quantities, the effect of trace components is less pronounced. However, when the ratio of CO:PO_3_G is 1:1, specifically under conditions of low II, trace fatty acids significantly alter polymer structure and affect the foam’s stability.

The hardness of eight samples (020, 022, 122, 200, 210, 212, 221, 231) fell in the required range of Shore A 15–22 hardness units. The hardness of the PB reference material also satisfied this requirement. The optimal values of PUF density are 0.2–0.3 kg/m^3^. As can be seen in Figure 6b, the density of 12 samples (011, 022, 102, 111, 120, 200, 201, 202, 210, 211, 221, 231) and the reference material fell within the required interval. Based on hardness and density, seven samples (022, 112, 200, 210, 212, 221, 231) were identified as having the potential to demonstrate superior performance and were taken forward for further analysis. PUF performance under applied static pressure was analyzed in the second step. Samples were studied at room temperature (23 °C) according to the SATRA TM 64 method and at an elevated temperature (40 °C). The results of the CS experiment are presented in Figure 6c. Although the link between composition and CS of the material is not clearly seen from the pattern, the trend between hardness and CS is observed. As can be concluded from Figure 6d, samples with lower hardness have a higher % of CS; the opposite trend is also true. Typically, all samples with Hardness Shore A 15–22, including the reference material, have CS lower than 10% under standard conditions. The CS properties of the synthesized materials as well as of the reference sample were studied. Seven samples (022, 200, 210, 212, 231, 112, 221) had CS values within the target range and were used in further testing.

Based on the SATRA TM6 standard, the water absorption of the insole material should be limited to 30%. According to Figure 6e, PUF samples are hydrophilic in most compositions. The samples with densities between 0.2–0.3 g/cm^3^ tend to have water absorption abilities within the required range. The water desorption abilities of all samples approached 100%, which is also a general requirement for insole materials. In conclusion, five samples (022, 200, 210, 212, and 231) demonstrated the required performance.

The value of tensile strength of Poron materials ranged from 0.9–1.2 MPa depending on thickness and design [12]. Hence, this range of properties is highly desirable for newly developed PUF samples. The samples with higher PO_3_G content (210, 212) had higher elasticity and could be stretched up to 450%. Sample 200, which is composed of a high level of PO_3_G:CO, had much lower elongation (226%) due to lower II and, therefore, lower HS concentration, which plays a significant role in the reinforcement of the structure. Furthermore, if II exceeds 100, for example in sample 231 where II is 140, tensile properties become affected by the high crystallinity of the polymer. The tensile characteristics of the samples are summarized in Table 6.

The tensile strength of the samples was lower than the optimal value of 1 MPa, though it approached the targeted value. The tensile strength can be improved by attaching textile support of adhesive backing to the insole, as in the case of the reference PB material.

### 3.4. Cushioning Properties

Both cushioning energy and cushioning factor characterize the ability of the insole to absorb shock waves resulting from foot strikes on hard surfaces. Estimation of cushioning properties is conducted through analysis of PUF compression curves, which is a better indicator of the insole’s functional performance than simple hardness or compression tests. The compression curves of the samples and the reference material are shown in Figure 6f.

As seen in Table 7, all samples had CE and CF values close to the targeted values and can be recommended for use as effective protective insoles during insole fabrication. The values of cushioning factor are slightly below the low thresholds for samples 210, 022 and PB. Low cushioning factor values are associated with stiffer materials. This may also be the case when the material has a textile cover (PB) or dense skin, as in the case of PUFs. The optimum thicknesses required for all samples to achieve a CE of 70 N·mm were calculated and are summarized in Table 7.

### 3.5. DSC Analysis

The degree of phase separation of PUF can also be studied using the DSC method. The thermogram of a pure PO_3_G polymer demonstrates typical transitions for amorphous PO_3_G polymers, including a step-like transition at −75.65 °C representing the glass transition temperature (T_g_), an exothermal peak at −54 °C representing cold temperature crystallization (T_c_) and endothermal peaks at 16–34 °C representing melting (T_m_) of a polydisperse polymer.

Characteristic PO_3_G transitions can be found in PUFs containing PO_3_G as part of SS. DSC thermograms of the PUFs are shown in Figure 7A. As can be seen from the figure, all samples have the T_g_ of SS characteristic for PO_3_G. In the case of samples 212, 231, 022 and 210, the temperature of glass transition has shifted toward positive values, implying that the movement of polyether chains is restricted, hence the domains are not well-organized in the PUFs. On the contrary, the T_g_ of sample 200 has slightly shifted towards the negative range, demonstrating that the polymer has a high degree of phase separation. Moreover, samples 200 and 210 display the other characteristic features of SS, i.e., cold temperature crystallization and melting temperatures. These PUFs have lower II than the remaining samples, hence less HS that restricts the ability of SS to crystallize.

Furthermore, the DSC thermogram demonstrates that samples 212, 210 and 231 are stable up to 250 °C, while samples 200 and 022 start to decompose at temperatures beyond 215 °C, which can be concluded from the presence of the endothermal shift in the DSC curve. Conclusions made based on DSC analysis are well aligned with the results of FTIR analysis of the PUFs (Figure 7B). The samples with higher degrees of phase separation have more pronounced adsorption bands corresponding to the H-bonded N-H group at 3305–3310 cm^−1^, hydrogen-bonded carbonyl groups of urea linkage at 1639 cm^−1^, and urethane linkage at 1694 cm^−1^.

In summary, Sample 200 is the most phase-separated, which may also explain its unique tensile characteristics and elasticity. On the other hand, samples 231, 022 and 212 are phase-mixed due to the presence of high amounts of HS in their structures.

### 3.6. Anaerobic Soil Biodegradation

Due to the molecular structure of CO, which contains polyester segments derived from vegetable oil, the polymeric surface is susceptible to attacks from microorganisms. Anaerobic soil biodegradation is close to environmentally benign conditions. The PUFs described in the present study combine two types of polyols: CO as a natural polyether and PO_3_G as a natural polyester. While the biodegradation of CO-based foams is a proven fact, polyester-based PUFs are known for their stability and much better resistance to hydrolysis. However, PO_3_G material, a corn-originated polyether, is known to be biodegradable.

SEM analysis showed that the samples had visual signs of biodegradation as well as characteristic spectral changes, deterioration of mechanical properties and weight loss. The foam underwent soil biodegradation and displayed surface morphological change, namely, white regions representing micro cracks and holes were observed. Optical microscopy images demonstrated the changes in cell strut morphology of PUFs before and after 2 and 4 months of soil biodegradation more vividly. Micro cracks were visible on the struts after 2 months of biodegradation and became more pronounced after 4 months in the soil media.

To substantiate the conclusions of the visual PUF examination, spectral changes were investigated. FTIR spectra of the reference foam before and after degradation are presented in Figure 8A–C. Distinct spectral changes are observed in the N-H region, C=O absorption band at 1643 cm^−1^ (strongly H-bonded, bidentate urea) loses intensity, C=O band at 1725 cm^−1^ shows a slight shift (to higher frequencies, 1735 cm^−1^) and broadening (Figure 8B) and a significant decrease in the band at 1537 cm^−1^ and 1237 cm^−1^ (Figure 8C).

The first visual signs of the physical deterioration (loss of transparency and formation of micro holes) of PUF foam may be related to the chain scission of the polyether/polyester soft segment and/or to the possible release of volatile compounds (micro holes). The development of micro-cracks and ruptures on the cell bun may be a consequence of the elimination of some hydrogen-bonding interactions between HS chains in addition to the previously mentioned molecular changes.

Figure 8D and Table 8 demonstrate that the samples lose elasticity after being exposed to soil media. According to this figure, the material becomes denser and more brittle after 2 months and undergoes further weakening after 4 months of biodegradation.

Changes in the weights of samples exposed to the soil environment are more pronounced than the change in the weight of the control. PUFs may lose weight due to the evaporation of residual water and emission of VOC. Total weight change after 4 months of the experiment reached only 2.99% on average. This important result demonstrates that the presence of renewably sourced polyether PO_3_G increases the stability of the PUF material; however, it remains susceptible to microbial biodegradation.

## 4. Conclusions

The present study demonstrated that the suggested formulation of PUF can be prepared using a combination of the pre-polymer method and heat curing. Characteristic parameters of PUF formulation such as track-free time and demold time are in the range of 18 ± 5.2 min, which makes the proposed formulation suitable for application using the RIM technique. Mechanical testing of the samples demonstrated that in most cases, PUFs with densities of 0.2–0.3 g/cm^3^ and hardness in the range of Shore A 15–22 had sufficient strength and weight-bearing characteristics. CS of the materials was as low as 0.5%, which is superior to current commercial materials (e.g., Plastozote^®^). Increasing temperature led to a slight softening of the foam and resulted in CS of 9%, on average, which is in the allowable range compared with the commercial insole material, Poron^®^ Blue.

As demonstrated by DSC analysis, the synthesized specimens have various morphologies depending on the composition. The samples with lower II and higher PO_3_G content (210 and 200) were found to have a well phase-separated structure, and the samples with higher II (and as a consequence higher HS content) were rather phase-mixed. Furthermore, the developed CO/PO_3_G/TDI formulations fall into the category of “green” biodegradable materials based on the results of degradability studies. Due to the presence of high amounts of polyether PO_3_G, the developed polyurethane formulations only display 2.99% weight loss on average after 4 months of the experiment. On the other hand, the mechanical properties of the samples were highly impacted.

Based on the outlined research results, several PUF samples satisfy the major requirements for quality-shock absorbing materials, i.e., the CE of the produced samples is higher than 70 N·mm and the CF of the produced samples is in the range of 4–8. This, in turn, ensures superior pressure distribution and shock adsorption, which are essential for stable functioning of FSR sensors. Considering the hygienic properties and dimensional stability of the foams after the application of steady loading at elevated temperatures, the CO/PO_3_G-based foams can be recommended for application in WBMS devices.

## Figures and Tables

**Figure 1 biomimetics-08-00334-f001:**
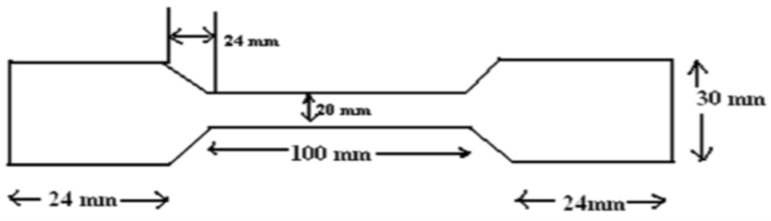
A dumb-bell-shaped specimen for testing tensile properties of insole materials based on the SATRA TM2 standard test method.

**Figure 2 biomimetics-08-00334-f002:**
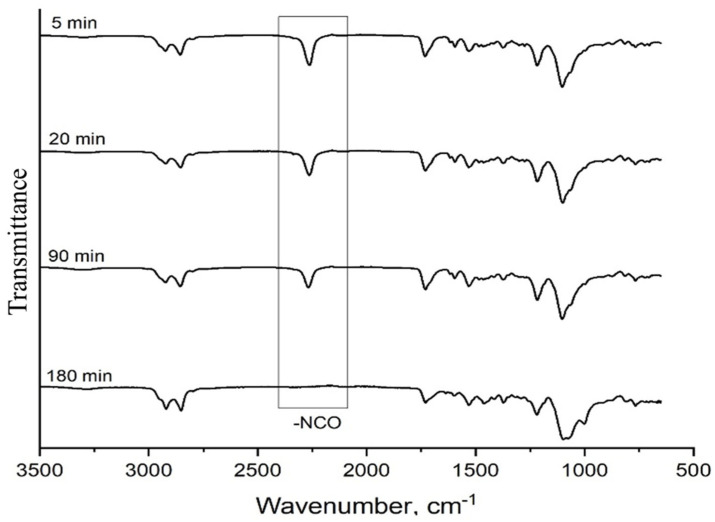
FTIR spectral change over the course of the polymerization reaction in the CO/PO_3_G/TDI foam sample (representative data for 1 pphp DBTDL and 0.1 pphp DABCO followed by post-curing).

**Figure 3 biomimetics-08-00334-f003:**
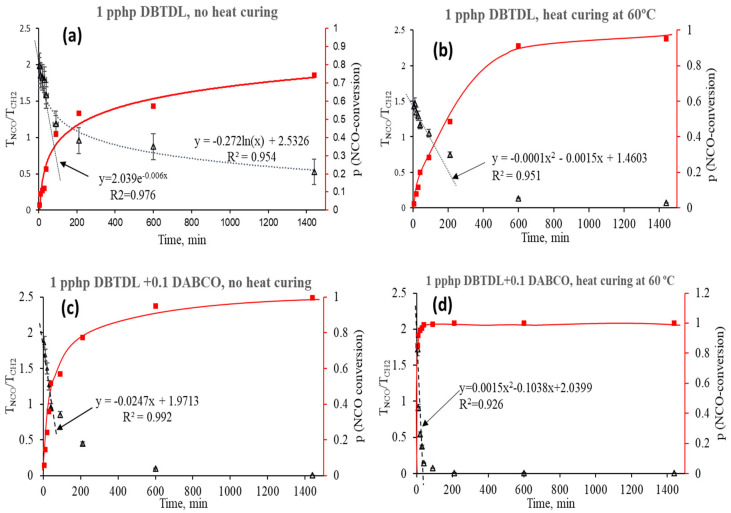
Kinetics of CO/PO3G/TDI foam synthesis under different experimental conditions: (**a**) 1 pphp DBTDL, no heat curing; (**b**) 1 pphp DBTDL, 0.1 pphp DABCO, no heat curing; (**c**) 1 pphp DBTDL followed by post-curing at 70 °C; (**d**) 1 pphp DBTDL and 0.1 pphp DABCO followed by post-curing.

**Figure 4 biomimetics-08-00334-f004:**
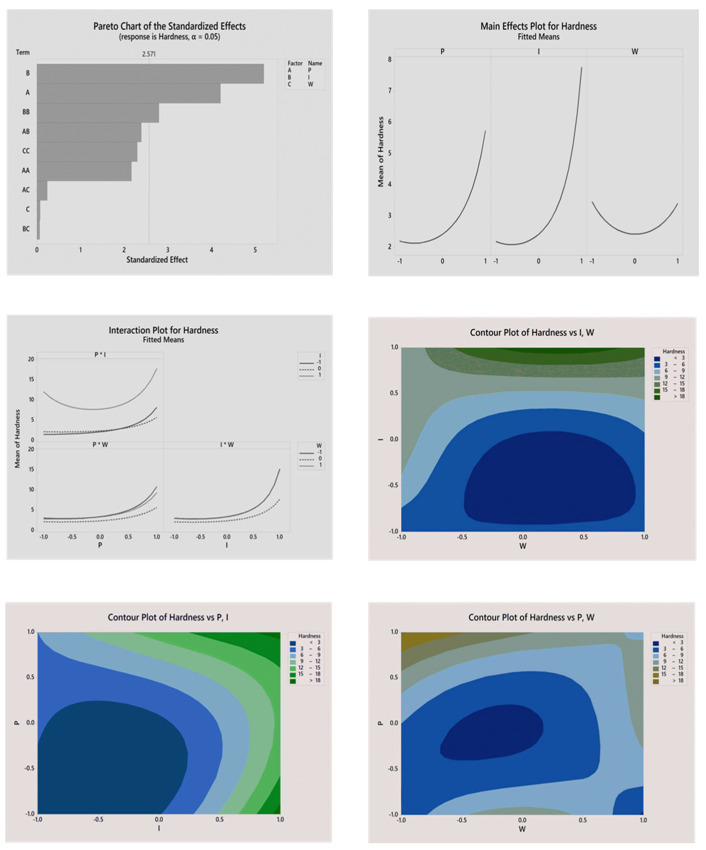
Statistical analysis of the impact of variables on the hardness of the material.

**Figure 5 biomimetics-08-00334-f005:**
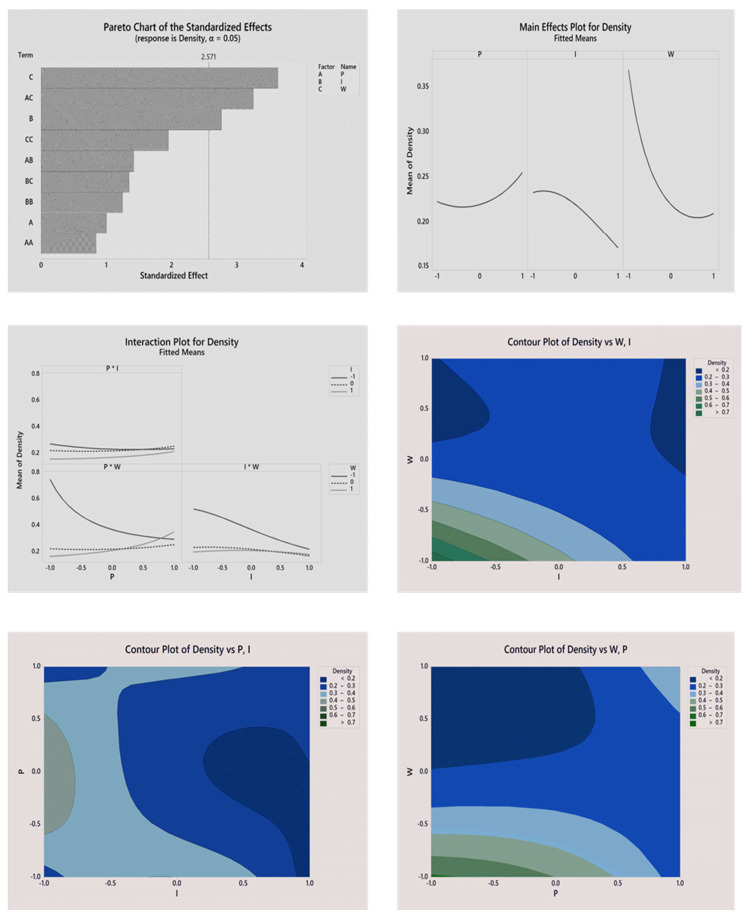
Statistical analysis of the impact of variables on the density of the material.

**Figure 6 biomimetics-08-00334-f006:**
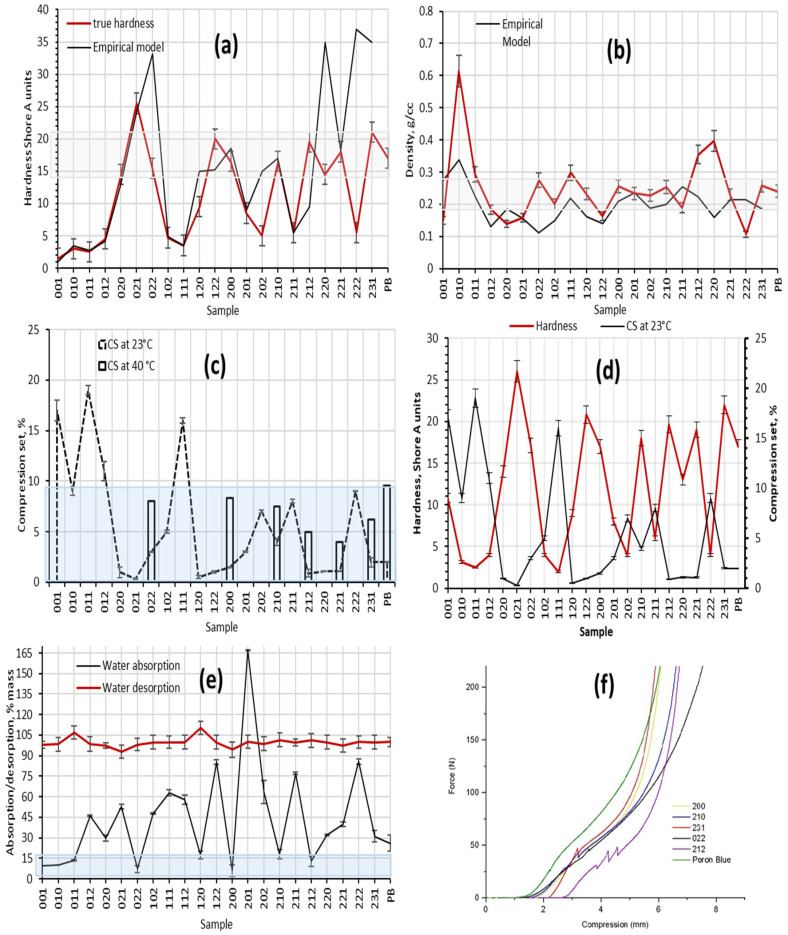
Results of mechanical and physical testing of PUF samples and PB-reference commercial material. (**a**) Hardness; (**b**) density; (**c**) Compression at different temperatures; (**d**) Hardness and CS; (**e**) Water absorption; (**f**) Compression curves.

**Figure 7 biomimetics-08-00334-f007:**
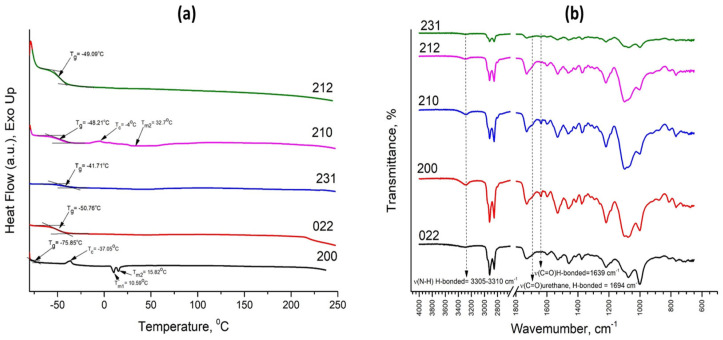
(**a**) DSC thermogram of the developed PUFS and (**b**) Changes in transmission in samples with various degrees of phase separation.

**Figure 8 biomimetics-08-00334-f008:**
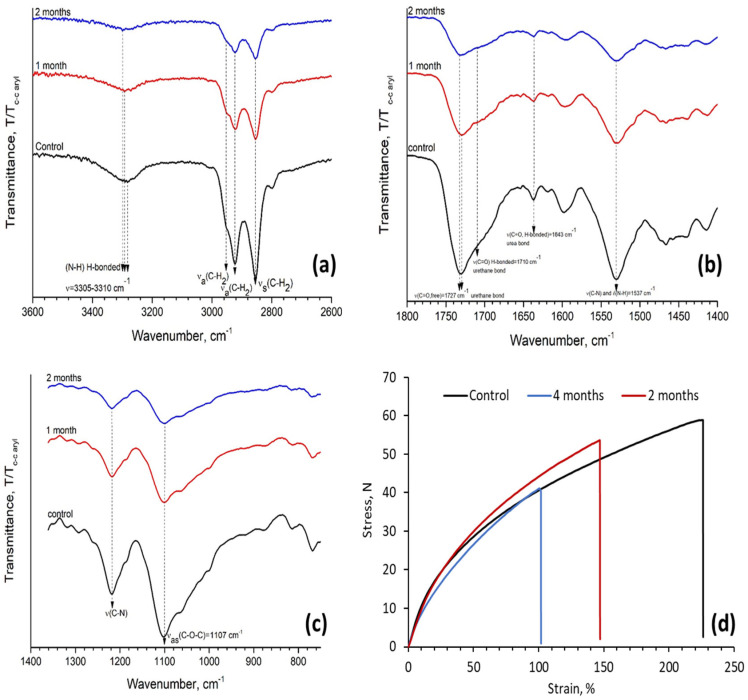
Spectral changes in N-H, C-H absorption region (**a**), C=O absorption (**b**), C-O-C (**c**) absorption regions of bio-degraded PUF foams and changes in mechanical properties of treated foam samples and control PUF. Sample 200 (**d**).

**Table 1 biomimetics-08-00334-t001:** Properties of raw materials (^a^ average molecular weight; ^b^ melting temperature; ^c^ functionality).

Role	Chemicals	Structure	M ^a^ [g/mol]	Physical State at T_ambient_	Tm ^b^[°C]	Density [g/cm^3^]	Fn ^c^
Isocyanate	TDI	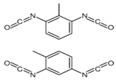	174.2	Clear liquid	20–22	1.225	2
Macrodiol	PO_3_G	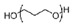	1043	Waxy solid	21.5	1.161	2
Crosslinker/ Chain extender	CO	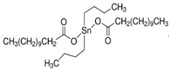	900	Viscous liquid	−18	9.61	2.7
Gelling catalyst	DBTDL	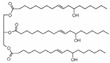	632	Clear liquid	10	1.066	-
Blowing catalyst	DABCO-33LV	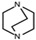	112.76	Clear liquid	-	1.2	2
Surfactant	NIAX L-580	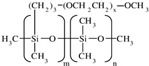	-	Viscous liquid	<0	1.030	-

**Table 2 biomimetics-08-00334-t002:** Box—Behnken response surface design matrix.

Run Order	Point Type	Blocks	Factors			Response Variable #1	Response Variable #2
			P	I	W	Density	Hardness
1	2	1	−1	−1	0	0.2522	1.5
2	2	1	−1	0	1	0.1834	2.5
3	2	1	−1	0	−1	0.6148	3.5
4	2	1	0	−1	1	0.1890	4.0
5	2	1	1	−1	0	0.2339	6.0
6	2	1	−1	1	0	0.1585	19.5
7	2	1	0	−1	−1	0.7665	3.0
8	2	1	0	1	−1	0.2319	9.0
9	2	1	1	0	−1	0.2542	18.5
10	0	1	0	0	0	0.2579	2.5
11	2	1	1	1	0	0.2303	19.0
12	2	1	0	1	1	0.1629	16.5
13	0	1	0	0	0	0.2751	2.0
14	0	1	0	0	0	0.2574	2.9
15	2	1	1	0	1	0.394494	8.5

**Table 3 biomimetics-08-00334-t003:** Characteristic production times for regimes A–D.

Regime	Cream Time [s]	Rise Time [s]	Gel Time [min]	Tack-Free Time [min]
A	296 ± 26	618 ± 29	42 ± 2	596 ± 8
B	43 ± 6	182 ± 12	6.33 ± 1	53.3 ± 6
C	13 ± 3	26 ± 2	19.3 ± 3	153 ± 14
D	10 ± 2	9 ± 1	1.5 ± 0.5	18 ± 5

**Table 4 biomimetics-08-00334-t004:** Analysis of Variance of transformed response.

Source	DF	Adj. SS	Adj. MS	F-Value	*p*-Value
Model	9	0.488857	0.054317	7.31	0.021
Linear	3	0.331741	0.110580	14.89	0.006
P	1	0.131139	0.131139	17.66	0.008
I	1	0.200569	0.200569	27.00	0.003
W	1	0.000034	0.000034	0.00	0.949
Square	3	0.114418	0.038139	5.13	0.055
P × P	1	0.034681	0.034681	4.67	0.083
I × I	1	0.057868	0.057868	7.79	0.038
W × W	1	0.039061	0.039061	5.26	0.070
2-Way Interaction	3	0.042698	0.014233	1.92	0.245
P × I	1	0.042273	0.042273	5.69	0.063
P × W	1	0.000401	0.000401	0.05	0.825
I × W	1	0.000024	0.000024	0.00	0.957
Error	5	0.037138	0.007428		
Lack-of-Fit	3	0.029807	0.009936	2.71	0.281
Pure Error	2	0.007331	0.003665		
Total	14	0.525995			

**Table 5 biomimetics-08-00334-t005:** Analysis of Variance of Transformed Response of density as a response variable.

Source	DF	Adj. SS	Adj. MS	F-Value	*p*-Value
Model	9	27.4443	3.0494	4.74	0.051
Linear	3	14.0312	4.6771	7.27	0.028
P	1	0.6424	0.6424	1.00	0.364
I	1	4.9058	4.9058	7.63	0.040
W	1	8.4830	8.4830	13.19	0.015
Square	3	4.1268	1.3756	2.14	0.214
P × P	1	0.4586	0.4586	0.71	0.437
I × I	1	1.0050	1.0050	1.56	0.267
W × W	1	2.4482	2.4482	3.81	0.109
2-Way Interaction	3	9.2863	3.0954	4.81	0.062
P × I	1	1.2950	1.2950	2.01	0.215
P × W	1	6.8237	6.8237	10.61	0.023
I × W	1	1.1677	1.1677	1.82	0.236
Error	5	3.2163	0.6433		
Lack-of-Fit	3	1.8264	0.6088	0.88	0.572
Pure Error	2	1.3899	0.6949		
Total	14	30.6607			

**Table 6 biomimetics-08-00334-t006:** Tensile characteristics of the PUFs and reference material.

Sample	Tensile Strength [MPa]	Tensile Strain [%]	Modulus of Elasticit [MPa]
022	0.09 ± 0.004	58.5 ± 7.05	0.11 ± 0.014
200	0.59 ± 0.032	226 ± 14.53	0.82 ± 0.015
210	0.36 ± 0.067	430 ± 8.97	0.28 ± 0.031
212	0.36 ± 0.120	444 ± 3.50	0.41 ± 0.050
231	0.58 ± 0.040	161.5 ± 5.93	0.99 ± 0.009
PB	1.83 ± 0.560	54 ± 3.00	3.7 ± 0.580

**Table 7 biomimetics-08-00334-t007:** Cushioning energy and cushioning factor of the developed PUFs and reference material.

Sample Thickness [mm]	Cushioning Energy [N·mm]		Cushioning Factor		Minimal Insole Thickness [mm]
	Running	Walking	Running	Walking	
200 (7 mm)	277 ± 13.89	177 ± 8.85	5.4 ± 0.27	4.4 ± 0.22	3.50
210 (7 mm)	336.9 ± 16.84	210.9 ± 10.54	4.4 ± 0.22	3.7 ± 0.23	2.20
022 (7.5 mm)	466 ± 23.33	229.3 ± 11.46	3.7 ± 0.18	3.9 ± 0.25	2.50
212 (7 mm)	243.7 ± 12.81	152.8 ± 7.64	6.2 ± 0.31	5.1 ± 0.26	3.50
231 (6.5 mm)	264 ± 13.20	373.8 ± 18.68	4.9 ± 0.24	4.0 ± 0.20	3.25
PB (6.5 mm)	373 ±18.69	373.8 ± 18.68	3.7 ± 0.24	3.7 ± 0.85	2.10
Targeted Value	Min. 100	Min. 70	4–8	4–8	-

**Table 8 biomimetics-08-00334-t008:** Tensile characteristics of the untreated control sample and the PUF samples after 2 and 4 months of anaerobic soil biodegradation.

Sample	Tensile Strength [MPa]	Tensile Strain [%]	Modulus of Elasticity [MPa]
Control	0.51 ± 0.004	210 ± 7.05	0.82 ± 0.140
2 months	0.41 ± 0.032	140 ± 14.53	0.69 ± 0.015
4 months	0.31 ± 0.067	93 ± 8.97	0.75 ± 0.031

## Data Availability

The data in this paper are shown in the graphs in the paper.

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
