# Peer review of "Biomimetic Orthopedic Footwear Advanced Insole Materials to Be Used in Medical Casts for Weight-Bearing Monitoring"

_biomimetics, 2023, doi:10.3390/biomimetics8040334_

Round 1

Reviewer 1 Report

In this work  renewably sourced Polyurethane Foam (PUF) made from poly(trimethylene ether) glycol (PO3G) and unmodified castor oil (CO) were synthesized and evaluated as  biomimetic orthopedic footwear advanced  insole materials.  Some issues need to be addressed in more depth:

-          Title – why “Biomimetic” term was used?

-          Abstract: “renewably sourced Polyurethane Foam (PUF) composed of poly(trimethylene ether) glycol (PO3G) and unmodified castor oil (CO)” – It is not a composite, so  the word “composed of” is inappropriate;

-          Table 1  - the structure of the chemicals cannot be  read; how the castor oil was characterized?

-          Fig. 2 – what is the description of OY axis?

-          What about the morphology of the foams, and the morphology-property relationships?

-          Line 606:  Due to presence of high amount of polyether PO3G, the developed polyurethane formulations only display 2.99% weight loss on average after 4 months of the experiment” –is this based on TG analysis?

Author Response

Reviewer 1

  1. Reviewer inquired about the reason for using the term “Biomimetic” in the title.

The meaning of Biomimetic Orthopedic Footwear is to refer to a designed footwear design that mimics properties of orthopedic footwear and in the same time can be utilized for the Weight-Bearing Monitoring application.

  1. Reviewer proposed that the use of the word “composed” in Abstract: “renewably sourced Polyurethane Foam (PUF) composed of poly(trimethylene ether) glycol (PO3G) and unmodified castor oil (CO)” is inappropriate.

We understand that “A composite material can be defined as a combination of two or more materials”, so accordingly we feel that the description can be used here.

  1. Table 1 - the structure of the chemicals cannot be read; how the castor oil was characterized?

All chemical structures were replaced with better quality ones in Table 1. Castor oil was purchased in purified form from Sigma Aldrich and was used as is.

  1. 2 – what is the description of Y axis?

Y-axis is the “Transmittance”; this was added to Figure 2 in the revised manuscript.

  1. What about the morphology of the foams, and the morphology-property relationships?

The morphological change and the morphology of the foams were characterized visually by SEM and by DSC analysis. These analyses were enough for the present work due to the interest in mimicking mechanical properties of orthopedic insoles.

  1. Line 606:  Due to presence of high amount of polyether PO3G, the developed polyurethane formulations only display 2.99% weight loss on average after 4 months of the experiment” –is this based on TG analysis?

Weight loss was quantified gravimetrically for all samples that were tested for biodegradation. Biodegradation tests were conducted following the “Soil Burial Method” under laboratory conditions in compliance with ASTM D 5988.

Reviewer 2 Report

The authors have done a lot of work on the production of polyurethane samples with different physical and mechanical characteristics.

However, there are a number of questions and comments.
1.
Table 1. It is necessary to redraw, preferably in special programs, the structural formulas of the used compounds.

2. The authors claim that « Hardness of the foam is also linked to isocyanate level, i.e., the higher the isocyanate index, the harder segments (HS) are formed in the polymer (hence harder the foam). Unfortunately, this is not so. Harder segments are not formed, but a greater number of hard segments are formed. I recommend that the authors supplement the literature review on the properties of polyurethanes. From recent works, for example [Slobodinyuk D. et al. Simple and Efficient Synthesis of Oligoetherdiamines: Hardeners of Epoxyurethane Oligomers for Obtaining Coatings with Shape Memory Effect //Polymers. – 2023. – Т. 15. – â„–. 11. – С. 2450]

3. According to what standard did the authors conduct the Compression Set? (ASTM?)

4. The authors claim that Con- 541 clusions made based on DSC analysis are well aligned with the results of FTIR analysis of the PUFs (Figure 7B). I would like to see explanations for Figure 7B. How the authors determined the degree of microphase separation by FTIR analysis? Unfortunately, the very low quality of the drawings in the article does not allow a more detailed understanding without a description.

5. The authors determine thermal effects in polyurethane samples at temperatures above 200 °C. It is strange that there are no exothermic effects of polymer oxidation by air oxygen. However, it is not indicated that the effects are determined in an inert atmosphere. Need to be clarified

6. Table 3: the time is measured very accurately, with an accuracy of 0.01 seconds, which causes some bewilderment. I suggest leaving only whole numbers. Similarly for other indicators in the table

After correcting these comments, I recommend accepting the article for publication.

Author Response

Reviewer 2

  1. Table 1. It is necessary to redraw, preferably in special programs, the structural formulas of the used compounds.

All structures were replaced with better quality structure drawings.

  1. According to what standard did the authors conduct the Compression Set? (ASTM?)

As mentioned under section 2.4.2.5, Tensile properties of insole materials were studied in compliance with SATRA TM 2 test method (Tensile properties of insole materials)..

  1. The authors claim that Con- 541 cushions made based on DSC analysis are well aligned with the results of FTIR analysis of the PUFs (Figure 7B). I would like to see explanations for Figure 7B. How the authors determined the degree of microphase separation by FTIR analysis? Unfortunately, the very low quality of the drawings in the article does not allow a more detailed understanding without a description.

The following explanation was added to the revised version of the manuscript: “Conclusions made based on DSC analysis are well aligned with the results of FTIR analysis of the PUFs. The samples with higher degree of phase separation have more pronounced adsorption bands corresponding to H-bonded N-H group at 3305-3310 cm-1, hydrogen bonded carbonyl groups of urea linkage at 1639 cm-1, and urethane linkage at 1694 cm-1.”.

We also agree with the reviewer about the bad quality of all Figures, and accordingly all figures’ quality was improved significantly in the revised manuscript.

  1. The authors determine thermal effects in polyurethane samples at temperatures above 200 °C. It is strange that there are no exothermic effects of polymer oxidation by air oxygen. However, it is not indicated that the effects are determined in an inert atmosphere. Need to be clarified

The objective of the study was to investigate the glass transition behavior and phase separation in the soft segments of polyurethane foams. The absence of exothermic peaks in the Differential Scanning Calorimetry (DSC) thermograms above 200 °C does not indicate the absence of polymer oxidation, but rather suggests the foams' stability in terms of oxidation within the tested temperature range. The primary focus was on detecting endothermic peaks associated with the glass transition, which confirmed the occurrence of phase separation and provided valuable insights into the material's structural properties.

  1. Table 3: the time is measured very accurately, with an accuracy of 0.01 seconds, which causes some bewilderment. I suggest leaving only whole numbers. Similarly, for other indicators in the table.

We agree with the reviewer’s comment. Numbers in Table 3 were rounded up to the closest integer.

Reviewer 3 Report

Dear Authors/Editor,

Thank You very much for the trust and opportunity to revise scientific paper entitled: “Biomimetic Orthopedic Footwear Advanced Insole Materials to Be Used in Medical Casts for Weight-Bearing Monitoringwritten by Sofya Rubtsova and Yaser Dahman for Your honourable Biomimetics journal.

In my opinion the article is quite interesting and consists a lot of interesting tests and results. The topic is also interesting especially for research groups connecting to polyurethanes and the application of new materials. Presented results can be valuable for modern PUR and a little “green” PUF materials and technology. The Interesting and useful thing is to use the design of experiment method to optimize the NCO: OH: blowing agent ratio for material designing with certain propertis. Testing methods and  results are well described. Conclusions are clear and quite well supported.

The introduction contain the big portion of information about should cushioning and sensor materials but the description and literature review about PU is rather poor. While the whole research is strongly oriented on polyurethane preparation. Moreover the list of  references are rather short and most of them are older than 10 years so I recommend to add some fresher positions.

So I recommend also to introduce the below listed corrections.

Page 1(Abstract),  lines 11 and 15,

The abbreviation “PO3G”should be the same not “PO3G”

Page 3 Table 1Table 1. Raw materials properties. a – number average molecular weight; b- melting temperature;c- functionality

The figures of chemical structure are mixed and not visible – it should be corrected!

Page 3, chapter 2.2. Preparation of PUFs, line 129 -131

There is only an information  about chemical reaction but the description of the chemical reaction is very short. More information, schematic reactions should be added.

After dot there is a extra “-“ (line 131)

Page 6, line 223

The grammar mistake: “…7 kg-force/cm2 is applied…” it should be “7 kg-force/cm2 was applied”

Page 18, lines 558-560

There is an information that the changes in cell strut morphology of PUF before and after 2 and 4 months of soil biodegradation. In my opinion the biodegradation time of material which will be applied in wet and warm environment (footwear) should be really reconsider. I think it should be more precisely described and discussed.

Page 20-21, References

The citation style should be the same according to the journal recommendation. For  example see: position 7 and 8 (line 637 and 640) as well as position 12 (line 647). It should corrected and please revise al this chapter.

In spite of the comments above I Generally recommend to publish this paper after proposed improvement.

Best regards

Author Response

Reviewer 3

The introduction contains the big portion of information about should cushioning and sensor materials but the description and literature review about PU is rather poor. While the whole research is strongly oriented on polyurethane preparation. Moreover, the list of references is rather short and most of them are older than 10 years so I recommend to add some fresher positions.

The focus of the study was primarily on determining optimal parameters for synthesizing insoles using polyurethane. The literature review was conducted with the aim of identifying these parameters and briefly analysing existing materials on the market, while also considering the concept of environmental sustainability in the literature. The utilization of PU was justified by the literature included in the original submission.

Page 1(Abstract), lines 11 and 15,

The abbreviation “PO3G” should be the same not “PO3G”.

This has been corrected throughout the manuscript.

Page 3 Table 1

The figures of chemical structure are mixed and not visible – it should be corrected!

All structural figures were replaced with better quality ones.

Page 3, chapter 2.2. Preparation of PUFs, line 129 -131

There is only an information about chemical reaction but the description of the chemical reaction is very short. More information, schematic reactions should be added.

As mentioned in this section, “Preparation of polyurethane foam is a complicated chemical process that includes two types of chemical reactions: gelling (formation of a polymer) and blowing (formation of voids)”. Accordingly, proposing any kind of reaction scheme can be challenging and misleading.

After dot there is an extra “-“(line 131). Fixed.

Page 6, line 223

The grammar mistake: “…7 kg-force/cm2 is applied…” it should be “7 kg-force/cm2 was applied”.

This has been corrected.

Page 18, lines 558-560

There is an information that the changes in cell strut morphology of PUF before and after 2 and 4 months of soil biodegradation. In my opinion the biodegradation time of material which will be applied in wet and warm environment (footwear) should be really reconsider. I think it should be more precisely described and discussed.

Biodegradation test were conducted following the “Soil Burial Method” under laboratory conditions in compliance with ASTM D 5988. ASTM D5988-18 is intended to be used with any material that does not restrict the growth of bacteria and fungus in the soil. This test technique evaluates the degree of aerobic biodegradation as soil is a valuable source of inoculum for testing the biodegradability of materials in the environment since it contains so many different species. We believe that this represents main parameters that impacts biodegradations of the insole materials.

Page 20-21, References

The citation style should be the same according to the journal recommendation. For example, see: position 7 and 8 (line 637 and 640) as well as position 12 (line 647). It should corrected and please revise al this chapter.

This section has been reviewed and improved.

Round 2

Reviewer 1 Report

Authors have addressed all the comments from reviewer and the revised manuscript can be published. 

Reviewer 2 Report

The author has improved the quality of the manuscript. Therefore, I would recommend this article for final publication. 

Reviewer 3 Report

Dear Authors,

The paper is satisfactorily corrected. I accept it in the present form.